# Effects of Multivalent BRD Vaccine Treatment and Temperament on Performance and Feeding Behavior Responses to a BVDV1b Challenge in Beef Steers

**DOI:** 10.3390/ani11072133

**Published:** 2021-07-19

**Authors:** Paul Smith, Gordon Carstens, Chase Runyan, Julia Ridpath, Jason Sawyer, Andy Herring

**Affiliations:** 1Philbro Animal Health Corporation, Teaneck, NJ 07666, USA; Paul.Smith@PAHC.com; 2Department of Animal Science, Texas A&M University, College Station, TX 77843, USA; Andy.Herring@exchange.tamu.edu; 3Department of Agriculture, Angelo State University, San Angelo, TX 76904, USA; Chase.A.Runyan@gmail.com; 4USDA-ARS NADC, Ames, IA 50010, USA; RidpathConsulting@gmail.com; 5King Ranch Institute for Ranch Management, Texas A&M University, Kingsville, TX 78363, USA; Jason.Sawyer@tamuk.edu

**Keywords:** bovine viral diarrhea virus, cattle respiratory vaccine, feeding behavior, feed intake, *Bos indicus* crossbred, performance

## Abstract

**Simple Summary:**

Bovine respiratory disease (BRD) threatens cattle production and welfare globally. We sought to quantify the effects of vaccine treatments and animal temperament classification on feed intake, feeding behavior and weight responses following challenge with bovine viral diarrhea virus, one of the pathogens associated with bovine respiratory disease. Commercially available respiratory vaccines were utilized, and temperament classification was based on exit velocity. Although overt clinical signs of respiratory disease were not observed following challenge, feed intake, weight gain, feed efficiency, and feed bunk frequency and duration were negatively affected. Animals administered a modified-live vaccine had more desirable feed intake, feed bunk duration, longer meal events and slower eating rates compared with those administered a killed or no vaccine. Temperament affected feeding behavior patterns, where calm steers had a greater duration of feed bunk visits and meal events, and slower eating rates compared with excitable steers. There were greater differences due to vaccine treatments in most feeding behavior traits within calm vs. excitable steers. The modified live vaccine mitigated the negative effects of the viral challenge to a greater extent than the killed vaccine for feed intake and feeding behavior patterns, and corresponded with previously reported findings regarding the effects of these vaccine types on immune responses.

**Abstract:**

This study examined the effects of multivalent respiratory vaccine treatment (VT) and animal temperament classification on feeding behavior traits, feed intake and animal performance in response to a bovine viral diarrhea virus (BVDV) challenge. Nellore–Angus crossbred steers (*n* = 360; initial body weight (BW) 330 ± 48 kg) were assigned to one of three vaccine treatments: non-vaccinated (NON), modified live (MLV) and killed (KV) regarding respiratory viral pathogens, and inoculated intranasally with the same BVDV1b strain. Cattle temperament categories were based on exit velocity. Overt clinical signs of respiratory disease were not observed, yet the frequency and duration of bunk visit events as well as traditional performance traits decreased (*p* < 0.01) following BVDV challenge and then rebounded in compensatory fashion. The reduction in dry matter intake (DMI) was less (*p* < 0.05) for MLV-vaccinated steers, and MLV-vaccinated steers had longer (*p* < 0.01) durations of bunk visit and meal events and slower (*p* < 0.01) eating rates compared with KV- and non-vaccinated steers following BVDV challenge. Greater differences in most feeding behavior traits due to VT existed within calm vs. excitable steers. Respiratory vaccination can reduce the sub-clinical feeding behavior and performance effects of BVDV in cattle, and the same impacts may not occur across all temperament categories.

## 1. Introduction

Although multivalent vaccines for bovine respiratory disease (BRD) are widely used in US feedlots [1], BRD remains one of the most costly and prevalent diseases in the beef industry [1,2,3,4]. The efficacy of BRD vaccines is impacted by a number of management factors, including stressors associated with weaning, commingling and transportation [5,6], as well as animal temperament [7,8].

Vaccine type also affects the degree of protection against BRD. In general, modified-live vaccines (MLV) have been shown to elicit more robust and longer-lasting immune responses [9,10] compared with killed vaccines (KV). Furthermore, modified live vaccination programs have been reported to reduce morbidity [11] and mortality rates [12], and lymphocytopenia [13] and pyrexia [14] compared with killed or no vaccine programs.

Cattle with more excitable temperaments have been reported to have compromised immune functions compared with calm cattle, which may be due to stress-induced responses associated with elevated serum cortisol concentrations [15,16]. Steers with excitable temperament phenotypes had lower in vivo lymphocyte proliferation and lower in vivo vaccine-specific IgG concentrations [7], and bulls with excitable temperaments have been shown to have reduced innate immune responses [17]. These results would suggest that temperament may alter the magnitude of protection against BRD elicited by vaccines.

There is limited research examining the effect of vaccine treatment (VT) and (or) the interaction of VT × temperament on feed intake, performance and feeding behavior responses in cattle following a disease challenge. Therefore, the objectives of this study were to examine the effects of multivalent VT for BRD and temperament classification on feed intake, performance and feeding behavior responses following a bovine viral diarrhea virus (BVDV) challenge in growing beef steers.

## 2. Materials and Methods

### 2.1. Animal and Experimental Design

All animal procedures were reviewed and approved by the Texas A&M University Institutional Animal Care and Use Committee as well as the Texas A&M University Institutional Biosafety Committee (Animal Use Protocols #2010-08 and #2013-0069). The animals utilized in this study were half-blood (F_2_ and F_3_) Angus–Nellore steers (*n* = 360) from the Texas A&M University McGregor Genomics herd, which consists of a *Bos taurus–Bos indicus* crossbred population that was specifically developed to support genomic studies. Four trials were conducted during consecutive years from 2010 to 2013. The steers were born in the spring (mid-February to late April annually) and were not vaccinated against BRD pathogens as calves. Steers were weaned at approximately 7 mo of age and received 3 clostridial vaccinations with Closti Shield 7 (Novartis Animal Health US, Inc., Greensbro, NC, USA) at approximately 70 day of age, at 3 weeks prior to weaning and at weaning. Following weaning each year, calves were managed as a single group and remained on pasture or were fed a growing ration, depending on the year, until being transported 165 km from McGregor, TX, to College Station, TX, in January or February. Steers were confirmed to be BVDV-PI negative through evaluation of ear-notch samples by antigen capture ELISA and were seronegative for BVDV antibodies (Texas Veterinary Medical Diagnostic Laboratory; TVMDL, Amarillo, TX, USA). Throughout this study, low-stress cattle handling methods were emphasized during movement, processing and data collection.

Animals were housed and managed as a single contemporary group each year following weaning until assignment to 1 of 4 pens at the Texas A&M University Beef Systems Research Unit (College Station, TX, USA). Each of the pens was equipped with 4 electronic feed bunks (GrowSafe System LTD., Airdrie, AB, Canada) with 20 to 26 steers per pen. A high-forage growing diet was used in this study that consisted of 31.5% corn, 36.5% chopped alfalfa, 24.5% dry distillers’ grains, 2.5% commercial premix and 5% molasses, which was formulated to meet the nutrient requirements of steers gaining 1 kg per day. The cattle were acclimated to the diet for 4 to 8 wk prior to the start of each year’s trial, and feed was delivered twice daily to ensure ab libitum access.

### 2.2. Vaccination and Challenge Protocols

At approximately 12 mo of age, steers were stratified by sire and genomic cow families, and randomly assigned to 1 of 3 vaccine treatments that consisted of a killed virus (KV) vaccine (*n* = 119), a modified live virus (MLV) vaccine (*n* = 123, and no (NON) vaccine (*n* = 118). Both vaccines were labeled for protection against infectious bovine rhinotracheitis, parainfluenza-3, bovine respiratory syncytial virus and bovine viral diarrhea, and were administered according to the label directions. Steers assigned to the KV treatment received an initial vaccine dose (Vira-shield; Novartis Animal Health US, Inc., Greensbro, NC, USA) 56 or 49 d prior to BVDV challenge, with a second dose administered 21 d later. Steers assigned to the MLV treatment were vaccinated with a single dose of Arsenal 4.1 (Novartis Animal Health US, Inc., Greensbro, NC, USA) on the same day that the second KV dose was administered. The non-vaccinated steers did not receive a BRD vaccine or sham injection prior to BVDV challenge, but the non-vaccinated steers were handled similarly each time to the KV- and MLV-vaccinated steers. The MLV-vaccinated steers were isolated from the KV- and non-vaccinated steers for 7 to 10 d following vaccination, with an empty pen between to avoid animal nose-to-nose contact and prevent potential cross-contamination from virus shedding. Following this post-vaccination isolation, steers were comingled again prior to being assigned to their respective study pens. The vaccine treatments were balanced across study pens.

All steers were challenged with the type 1b non-cytopathic BVDV strain CA0401186a that was obtained from the USDA-ARS National Animal Disease Center, Ames, IA [18]. This BVDV strain, originally isolated from a persistently infected BVDV Holstein calf, was selected for this study, as previous research demonstrated that it elicited typical immunological and clinical symptoms of morbidity, but with minimal risks of extreme illness or death [18], as is typical for most field strains of BVDV. Each steer was administered 5 mL of BVDV inoculum containing 1 × 10^5^ TCID/mL intranasally (2.5 mL dose per nasal passage). Challenge dates (Day 0) were 11 May, 10 May, 15 May and 4 June for trial years 2010–2013, respectively.

### 2.3. Data Collection

Body weight and rectal temperature were measured on Days –28, 0, 3, 7, 10, 14, 28 and 42 relative to the BVDV challenge (Day 0). Exit velocity was measured using infrared sensors on Days 0 and 14 as the time to transverse a fixed distance of 1.8 m upon exiting a squeeze chute (Farm Tec, Inc. North Wylie, TX, USA). Relative exit velocity (REV) was computed as (individual EV—mean EV) ÷ mean EV for each animal within year, and averaged for Days 0 and 14. Dry matter intake, ADG and feeding behavior traits were evaluated during each of the 4 14-d experimental periods (EP) relative to the BVDV challenge, which occurred on Day 0: Period 1 (Days –14 to −1), Period 2 (Days 0 to 13), Period 3 (Days 14 to 27) and Period 4 (Days 28 to 41). As BW was not measured on Day –14, ADG during the first 14-d period was computed from the BW measured on Days –28 and 0, with the assumption that growth was linear during the initial 28-d period. A timeline of the experimental procedures is provided in Figure 1.

The steers were observed twice daily during the first 14 d following the BVDV challenge, and once per day thereafter to assess the clinical symptoms associated with BRD. Evaluations of cough, ocular and nasal secretion, depression, diarrhea and anorexia were recorded using a 0 to 5 clinical illness score (CIS; 0 = no symptoms; 1 to 5 were indicative of least severe to most severe). The criterion used to define BRD cases in this study were clinical scores of >3 for a single clinical symptom or combined scores of ≥3 for 2 or more clinical symptoms. Rectal temperatures were recorded on pre-determined days rather than as a final clinical threshold following the initial clinical assessment, as in field protocols for BRD diagnosis [19]. Animals that exhibited a rectal temperature of >40 °C were administered tulathromycin (Draxxin, Zoetis Animal Health), regardless of their CIS. This manuscript is part of a series that reports on various animal responses to a BVDV challenge. Investigations of clinical symptoms [19] and sire effects on DMI and ADG [20] following BVDV challenge in these cattle have been previously reported. This manuscript focused on the effects of animal temperament and VT on DMI and feeding behavior responses following a BVDV challenge.

### 2.4. GrowSafe Data

A GrowSafe system (DAQ 6000E) was used to measure feed intake and feeding behavior traits from 14 d prior to until 42 d following the BVDV challenge. The system consisted of feed bunks equipped with load bars to measure feed disappearance and RFID antennas within each feed bunk to record animal presence via detection of EID ear tags. Assigned feed disappearance (AFD) rates were computed daily for each feed bunk to assess data quality. Data for each pen were omitted from analysis due to system malfunctions, power outages or low (<95%) pen average AFD rates. During the 2010 and 2013 trials, data for 14 d and 2 d, respectively, were removed due to low AFD rates. The average AFD for the remaining days were 97.1% and 99.3%, respectively. No data were removed from the 2011 and 2012 trials, with average AFD rates exceeding 99%.

The feeding behavior traits evaluated in this study were based on frequency and duration of bunk visit (BV) events, head-down (HD) duration, frequency and duration of meals events, and time to approach feed bunk (TTB) following feed-truck delivery (Table 1). A BV event commenced when the EID ear tag of an animal was first detected at the feed bunk and ended when the time between the last 2 consecutive EID recordings exceeded 100 s, the EID ear tag was detected at another feed bunk or the EID ear tag of another animal was detected at the same feed bunk [21]. Bunk visit frequency was defined as the number of independent events recorded, regardless of whether or not feed was consumed, and BV duration was defined as the sum lengths of all BV events recorded during a 24-h period [22]. Head-down duration was computed as the sum of the number of times an EID ear tag was detected each day multiplied by the scan rate of the GrowSafe system. R statistical software (R Core Team, 2014, Vienna, Austria) was used to compute TTB each day as the interval length between feed delivery for each pen and each animal’s first BV event following feed delivery [22]. Estimated values for missing feed intake data were derived from a linear regression of the feed intake on the day of the trial [23]. Bunk visit eating rate was computed as the ratio of daily DMI to daily BV duration.

To compute meal data, a 2-pool Gaussian–Weibull distribution model was fitted to log-transformed non-feeding interval data. The intercept of the 2 distributions was used to define the meal criterion [24,25], which was the longest non-feeding interval considered to part of a meal event. The individual animal meal criterion was used to compute the frequency and duration of daily meal events. Meal eating rate was computed as the ratio of daily DMI and daily meal duration.

### 2.5. Statistical Analysis

Mixed model procedures of SAS 9.4 (SAS Inst. Inc., Cary, NC, USA) were used to analyze DMI, ADG and feeding behavior data. The model included VT and EP as fixed effects; REV as a covariate; the interactions of VT × EP, VT × REV, EP × REV and VT × EP × REV; and the random effects of year and pen within year. The 3-way and the EP × REV interactions were non-significant for all dependent variables and thus were removed from the final models. Previous analyses of these data have documented significant sire effects on DMI and ADG [20]. However, in the current study, sire was excluded from the statistical models in order to fully evaluate the effects of temperament on the response variables, as sire differences in animal temperament have been demonstrated to exist in this population [26].

To examine the interactive effects between VT and REV, an unequal slope model was fitted for the dependent variables with significant (*p* < 0.05) VT × REV interactions. Subclass means for steers with calm and excitable temperaments were compared at mean REV minus 1 SD and mean REV plus 1 SD, respectively, using the PDIFF option in SAS. Differences between VT and EP least squares means were compared using the PDIFF option of SAS (SAS 9.4). Additionally, contrast statements were used to examine the nature of the dependent variable responses (linear, quadratic or cubic) across EP. Finally, to determine if temperament phenotype was equally distributed across the VT groups, steers were sorted by REV and classified into calm, moderate or excitable temperament groups based on them being ± 0.5 SD from the mean REV within the trial. The distribution of temperament classification within VT was examined with the PROC FREQ procedure of SAS using the CHISQ option (SAS 9.4).

## 3. Results

### 3.1. Vaccine Treatment and Experimental Period

The least squares means for DMI, performance and feeding behavior responses are presented in Table 2. Compared with Period 1, DMI, ADG and G:F were reduced by 15.9%, 27.7% and 20.0%, respectively, during the 14-d period following BVDV challenge (Period 2), and subsequently increased during Periods 3 and 4 in a cubic (*p* < 0.01) manner. Although the main effect of VT did not affect DMI, ADG or G:F, there was a VT × EP interaction (*p* < 0.05, Figure 2) for DMI. The reduction in DMI during Period 2 following BVDV challenge was less (*p* < 0.05) for MLV- (−10.6%) compared with KV- (−18.2%) and non-vaccinated steers (−18.9%); correspondingly, the subsequent increase in DMI during Period 3 was greater for KV- and non-vaccinated steers compared with MLV-vaccinated steers. While the VT × EP interaction was not significant (*p* = 0.11) for ADG, the effect of VT treatment on the reduction in ADG during Period 2 showed a similar trend to DMI (Figure 2).

Compared with Period 1, the frequency and duration of BV events, the frequency and duration of meal events, HD duration and the frequency of BV events per meal all decreased by 15% to 25% during Period 2 and subsequently increased during Period 3 in a cubic (*p* < 0.01) manner. In contrast to these feeding behavior traits, meal eating rate actually increased by 6.4% during the 14 d after the BVDV challenge (Period 2) and continued to increase during Period 4 in a cubic (*p* < 0.01) manner. In contrast to meal eating rate, eating rate during BV events was not affected by the BVDV challenge. Compared with Period 1, TTB following feed delivery increased (*p* < 0.05) by 37.1% during the 14 d after the BVDV challenge, with TTB decreasing during Period 3 and 4 to values similar to those observed prior to BVDV challenge.

In contrast to DMI, the VT × EP interactions were not significant for the feeding behavior traits. However, VT significantly altered feeding behavior traits throughout the study (Table 2), with MLV-vaccinated steers having distinctly different feeding behavior patterns compared with KV- and non-vaccinated steers. The MLV-vaccinated steers had 5% to 7% greater (*p* < 0.01) HD duration and durations of BV and meal events, and 4% to 5% slower (*p* < 0.01) BV and meal eating rates compared with KV- and non-vaccinated steers. Additionally, MLV steers had a 4% to 6% greater (*p* < 0.01) number of BV events per meal compared with KV- and non-vaccinated steers. The TTB following feed delivery and BV frequency were not affected by VT.

### 3.2. Temperament

The distribution of steers with divergent phenotypes for temperament across VT was evaluated based on being ± 0.5 SD from the mean REV of 0 ± 0.24. For steers with excitable temperaments, the distribution of steers was similar (*p* = 0.14) for NON, KV and MLV vaccine treatments at 28.2%, 33.3% and 38.5%, respectively. Likewise, steers with calm temperaments were equally distributed for NON, KV and MLV vaccine treatments (37.6%, 25.7% and 36.6%, respectively).

Relative exit velocity was a significant covariate (*p* < 0.01) for DMI, such that steers with calm temperaments (mean REV–1 SD) consumed 5.0% more feed than steers with excitable temperaments (mean REV + 1 SD), irrespective of the vaccine treatment (Table 3). There was a tendency (*p* = 0.08) for REV to affect ADG, with calm steers having a 5.3% numerically greater ADG compared with excitable steers. However, REV did not affect (*p* = 0.69) G:F, and the VT × REV interactions were not detected (*p* ≥ 0.29) for DMI, ADG or G:F.

With the exception of BV frequency and meal eating rate, REV was a significant covariate for all feeding behavior traits. In general, HD duration, and BV and meal duration all decreased (*p* < 0.01) as REV increased (Table 3). However, VT × REV interactions were detected (*p* < 0.05) for both HD and meal duration. In KV- and non-vaccinated steers, these traits were not affected by REV (the slopes did not differ from zero), but in MLV-vaccinated steers, both HD and meal duration decreased as initial REV increased (Figure 3). Within calm steers, MLV-vaccinated steers had greater (*p* < 0.01) HD and meal duration compared with KV- and non-vaccinated steers, whereas VT differences in HD and meal duration were not detected in steers with excitable temperaments. There were also significant VT × REV interactions for both BV and meal eating rates (Figure 3). Within steers with a calm temperament, MLV-vaccinated steers had lower BV and meal eating rates compared with KV- and non-vaccinated steers. However, BV and meal eating rates were not affected by VT in steers with excitable temperament phenotypes.

Although the frequency of BV events was not affected by REV, there was a VT × REV interaction (*p* < 0.05) for the frequency of meal events. Meal frequency increased as REV increased in KV- and non-vaccinated steers; however, REV had no effect on meal frequency in MLV-vaccinated steers (Figure 4). The KV- and non-vaccinated steers with excitable temperaments consumed more meals per day than MLV-vaccinated steers, whereas VT did not affect meal frequency in calm steers. Reflecting the influence of temperament on meal frequency, the number of BV events per meal declined (*p* < 0.05) as REV increased. Irrespective of VT, TTB following feed delivery was affected by REV, with excitable steers taking almost 4 min longer to approach the feed bunk following feed delivery than steers with calm temperaments (Table 3).

## 4. Discussion

### 4.1. Responses to the BVDV Challenge

This study demonstrated the effects of sub-clinical illness on performance traits in *Bos indicus* crossbred steers that Griffin [27] and others have reported as plaguing the beef industry. Detailed descriptions of health traits and antibody titer responses were previously reported by Downey-Slinker et al. [19]. In this study, 14% of the steers had a CIS of 1 or 2 (CIS = 0 to 5); however, none of the steers met the criteria for clinical BRD diagnosis following the BVDV challenge and none of the steers died. As reported by Downey-Slinker et al. [19], within the 14-d period following BVDV challenge, 40% of steers presented with pyrexia (1 SD over the Day 0 rectal temperature for two measurement days), 55% presented with lymphopenia (>40% reduction in lymphocyte counts) and 41% presented with thrombocytopenia (>40% reduction in platelet counts). Both lymphopenia and thrombocytopenia are well-established indicators of subclinical BVDV infection in beef cattle [12,13]. Other studies have also shown that animals challenged with BVDV Type 1b [10] or BVDV Type 2 [28] strains do not always manifest with observed clinical signs of BRD. Burciaga-Robles et al. [29] reported that calves challenged with BVDV Type 1b had minimal or no observed clinical signs of BRD.

Despite the lack of clinically diagnosed BRD cases, there were substantial reductions in DMI, ADG, G:F and feeding behavior traits during the 14-d period following the BVDV challenge. Similar patterns have been documented in other studies with clinically ill cattle. During a spontaneous outbreak of BRD in growing bulls (8–9 mo of age), Jackson et al. [22] reported that DMI was reduced by 39% during the week prior to observed clinical BRD diagnosis. Likewise, the frequency and duration of BV events declined by 2.9 events/d and 4.4 min/d, respectively, during the week prior to an observed clinical diagnosis of BRD [22]. Carlos-Valdez et al. [30] reported that Angus crossbred steers challenged with *Mannheimia haemolytica* after exposure to a persistently infected BVDV Type 1 calf had reduced DMI, ADG and G:F during the first 4 d following challenge. Similarly, Theurer et al. [31] reported that calves challenged with *M. haemolytica* spent less time at both the feed bunk and hay feeder compared with calves that were not challenged. In addition, Wolfger et al. [32] reported that an increase in feed intake per meal event, along with increases in the frequency and duration of meal events, was associated with a lower risk of developing BRD. Hutcheson and Cole [33] reported that calves observed to be clinically ill had 11% lower intake and 29% lower ADG compared with calves observed to be healthy. Sowell et al. [34] found that morbid steers had fewer feeding bouts and spent less time at the feed bunk compared with healthy steers, and Sowell et al. [35] reported that clinically healthy steers had more rapid responses following feed delivery than steers identified as being clinically ill. Likewise, Daniels et al. [36] reported that clinically ill calves had a lower frequency of feeding bouts and spent less time at the feed bunk compared with clinically healthy calves. In dairy cows, reductions in DMI and feeding behavior responses prior to and following a diagnosis of clinical mastitis [37], metritis [38] and ketosis [39,40] have been observed.

The effects of the BVDV challenge caused ADG, DMI and most of the feeding behavior traits to decline during the 14-d period immediately following the BVDV challenge (Period 2), which subsequently increased during Periods 3 and 4 in a cubic manner. These performance trait responses during Period 2 were associated with substantial reductions in lymphocyte and platelet counts on Day 14 following the BVDV challenge, as previously reported by Downey-Slinker et al. [19]. Compared with clinically healthy calves, Buhman et al. [41] reported that the frequency and duration of feeding events were reduced in morbid calves 11 to 27 d after feedlot arrival, and increased thereafter during the next 28-d period. These authors attributed this increase in feeding activity to a post-sickness compensation. Carlos-Valdez et al. [30] reported a post-sickness compensation in calves exposed to persistently infected BVDV calves for 72 h then challenged with *M. haemolytica.* Following reductions in DMI, ADG and G:F during Days 0 to 4 after *M. haemolytica* challenge compared with control calves, there was a subsequent increase in ADG and G:F during Days 5 to 17 following the challenge [30]. Calves challenged with *M. haemolytica* during Days 5 to 17 appeared to compensate for the loss in production and showed an increase in ADG and G:F compared with control calves [30]. Holland et al. [42] reported that crossbred heifers treated for BRD had a lower ADG compared with those not treated during the preconditioning phase; additionally, there was a greater compensation in ADG during the first 28 d following the preconditioning phase for cattle treated three times compared to cattle that had never been treated for BRD. A similar compensation was observed in this study, with DMI, ADG and G:F being substantially greater during Period 3 following the BVDV challenge. Likewise, the frequency and duration of BV events, and HD and meal durations were greater, and the TTB following feed delivery was faster during Period 3 than Period 2, demonstrating that the steers quickly compensated for the BVDV challenge in this study.

### 4.2. Vaccine Treatment Effects

The reduction in DMI following the BVDV challenge was less pronounced in MLV-vaccinated steers compared with the KV- and non-vaccinated steers. These results coincided with the previous findings of Downey-Slinker et al. [19], whereby MLV-vaccinated steers had a reduced (33.9%) incidence of lymphopenia compared with KV- (64.7%) and non-vaccinated steers (68.1%). Although VT did not affect the proportion of steers exhibiting pyrexia during the 14-d post-BVDV challenge period, the MLV-vaccinated steers had lower rectal temperatures compared with KV- and non-vaccinated steers on Days 3 and 7 following the challenge [19].

Vaccine treatment clearly altered feeding behavior patterns, such that MLV-vaccinated steers had a greater duration of both BV and meal events, greater HD duration and more BV events per meal compared with KV- and non-vaccinated steers. Additionally, as VT did not alter DMI, eating rates during both BV and meal events were slower in MLV- compared with KV- and non-vaccinated steers. These results, in conjunction with those of Downey-Slinker et al. [19], suggest that the multivalent MLV provided a greater level of protection to the BVDV challenge compared with the KV. Although some studies have reported no difference in antibody response between KV and MLV vaccines [43], Downey-Slinker et al. [19] found that the MLV-vaccinated steers in the present study had greater BVDV Type 1b titer concentrations compared with KV-vaccinated steers prior to BVDV challenge, but KV-vaccinated steers had greater titers at 14 d following the challenge. Additionally, MLV vaccines have been shown to reduce susceptibility to lymphopenia and reduce the fever response to a greater extent as compared with KV vaccines [13,14]. Collectively, results from these studies suggest that the MLV vaccine was more effective at mitigating subclinical symptoms of BRD compared with the KV vaccine.

### 4.3. Temperament Effects

The effects of temperament on the DMI and performance of cattle in multiple breeds have been well documented, such that more excitable steers have decreased DMI and ADG compared with calm steers [44,45,46]. In agreement with previous research, the results from the current study found that calm steers had greater DMI and numerically greater ADG compared with excitable steers. However, there have been mixed results on the effect of temperament on feed efficiency. Bruno et al. [45] reported that temperament did not affect G:F, even though cattle with calm temperaments had increased DMI and ADG compared with excitable cattle. *Bos indicus* crossbred steers [47] and heifers [46] with excitable temperament phenotypes had lower ADG and less favorable G:F compared with cattle with calm temperaments. Likewise, Cafe et al. [48] reported that Angus steers with excitable temperaments upon feedlot arrival tended to have less favorable G:F than steers with excitable temperaments. In contrast to the latter studies, temperament phenotype did not affect G:F in the current study among these 50% *Bos indicus* steers.

Reflecting the effect of temperament on DMI, steers with calm temperaments spent more time each day consuming feed, whether quantified as HD duration or the durations of BV or meal events. Furthermore, eating rate during BV events was slower in calm than excitable steers. Similarly, Cafe et al. [48] reported that cattle with a faster EV spent less time at the feed bunk compared with cattle with a slower EV. Olson et al. [46] reported that heifers with calm phenotypes had 9% greater meal duration, and consumed meals that were 22% longer and 17% larger compared with excitable heifers. These results, along with the results of Cafe et al. [48] and Olson et al. [46], suggest that cattle with calm temperaments have more favorable feeding behavior patterns compared with cattle with more excitable temperaments.

As previously discussed, the reduction in DMI associated with an increase in REV was not affected by VT. In contrast, the effects of temperament on feeding behavior responses were impacted by VT, suggesting that feeding behavior responses were more sensitive in detecting differences due to VT than DMI responses. The MLV-vaccinated steers with calm temperaments had greater HD and meal durations, and slower meal eating rates than KV- and non-vaccinated steers, whereas VT differences in these feeding behavior traits were not detected in steers with excitable temperaments.

It is interesting to note that the MLV-vaccinated steers exhibited favorable feeding behavior patterns, despite the fact that there was a numerically higher proportion of steers with excitable temperaments in the MLV- compared with the non- and KV-vaccinated treatments (38.5%, 28.2% and 33.3%, respectively). The results from this study would suggest that the benefits of the multivalent MLV vaccine were more evident in calm than in excitable steers, which may be related to the fact that steers with an excitable temperament have heightened physiological responses to stress. Multiple studies have reported that excitable steers have greater cortisol responses to stress than calm steers [16], which has been shown to negatively affect the immunocompetence of cattle [49]. In addition, Oliphint et al. [7] reported that cattle with excitable temperaments had reduced immune responses to vaccination compared with calm cattle. These results suggest the potential beneficial effects of the MLV vaccine may have been mitigated by the increased stress responsiveness exhibited by excitable temperaments. Furthermore, previous analyses of data from this study demonstrated that subjective temperament scores were moderately correlated in a negative manner with antibody titer responses following the BVDV challenge [50].

## 5. Conclusions

The objectives of this study were to quantify the effects of the multivalent VT and animal temperament on DMI, performance and feeding behavior responses following a BVDV1b challenge. The results demonstrated that the multivalent VT clearly altered DMI following BVDV challenge, such that MLV-vaccinated steers had less of a reduction in DMI compared with KV- and non-vaccinated steers. Furthermore, feeding behavior patterns were substantially affected by VT, with MLV-vaccinated steers having increased feeding duration and slower eating rates compared with KV- and non-vaccinated steers. These results, in conjunction with those of Downey-Slinker et al. [19], suggested that the MLV vaccine mitigated the impacts of the BVDV challenge to a greater extent compared with the KV and NON treatments. Additionally, temperament affected DMI and feeding behavior patterns, with calm steers having increased DMI and feeding durations, and slower eating rates during the BV events compared with excitable steers. Previous analyses in these cattle have demonstrated a substantial genetic influence for temperament at weaning [26], and for DMI and ADG following this BVDV challenge [20]. We observed that the same impacts of VT may not occur across all animal temperament categories The increased stress responsiveness of excitable steers in this study appeared to have mitigated the beneficial effects of the MLV vaccine. This study indicated that temperament classification may be at least a partial proxy for genetic background when pedigrees are unavailable.

## Figures and Tables

**Figure 1 animals-11-02133-f001:**
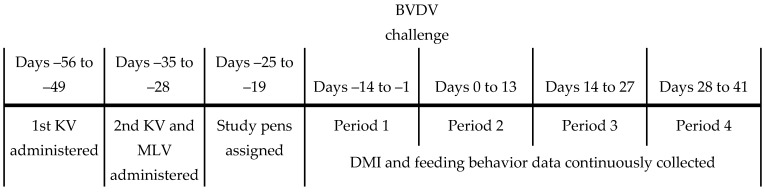
Timeline of experimental procedures during the study. Steers were housed as a single group each year until the modified live virus (MLV) vaccine was administered. Thereafter, the MLV-vaccinated steers were isolated from the killed virus (KV) and non-vaccinated (NON) steers for 7 to 10 d, and then comingled again until being assigned to their study pens 5 to 10 d before the start of Period 1. All steers were subjected to the BVDV challenge on Day 0.

**Figure 2 animals-11-02133-f002:**
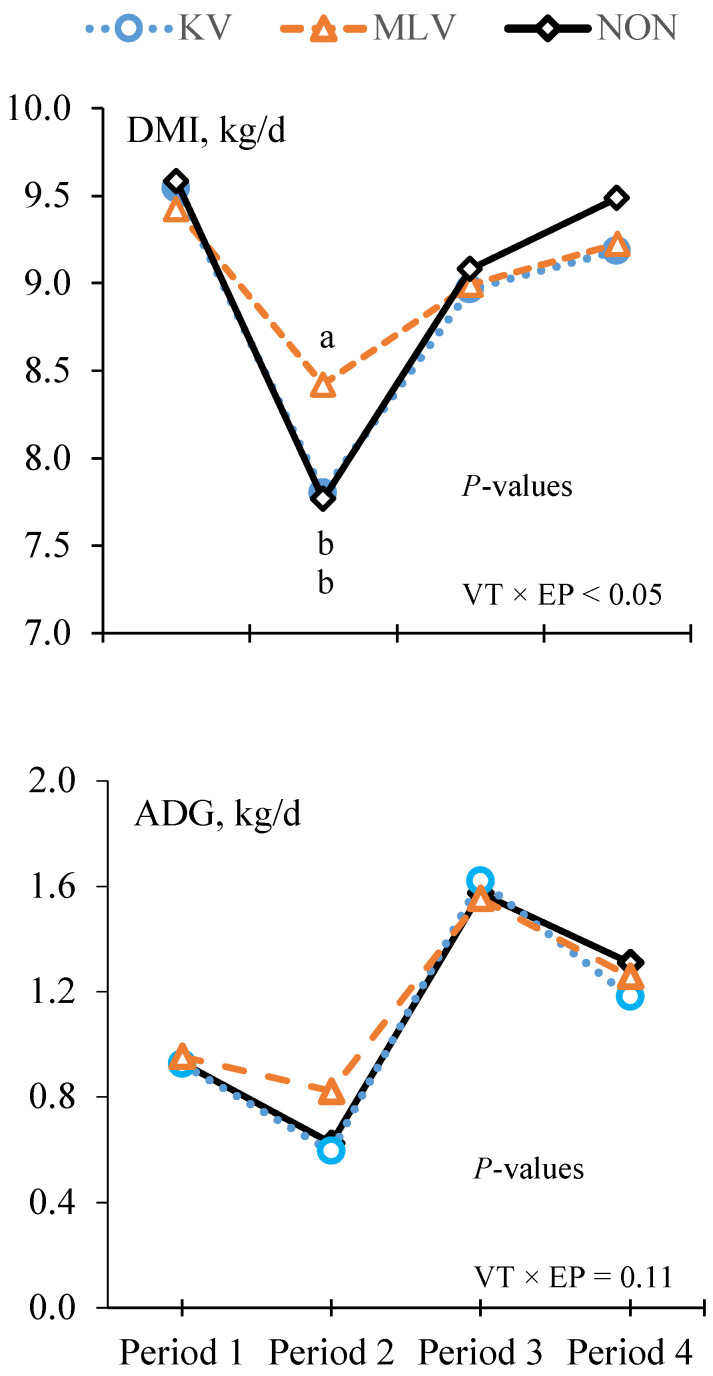
Effects of vaccine treatment (VT) and experimental period (EP) on DMI and ADG. ^a,b^ DMI differed (*p* < 0.05) between subclass means. Vaccine treatments include modified live vaccine (MLV; *n* = 123), non-vaccinated (NON; *n* = 118) and killed vaccine (KV; *n* = 119) steers. Experimental periods correspond to the 14-d intervals before (Period 1) and following the BVDV challenge (Periods 2, 3 and 4).

**Figure 3 animals-11-02133-f003:**
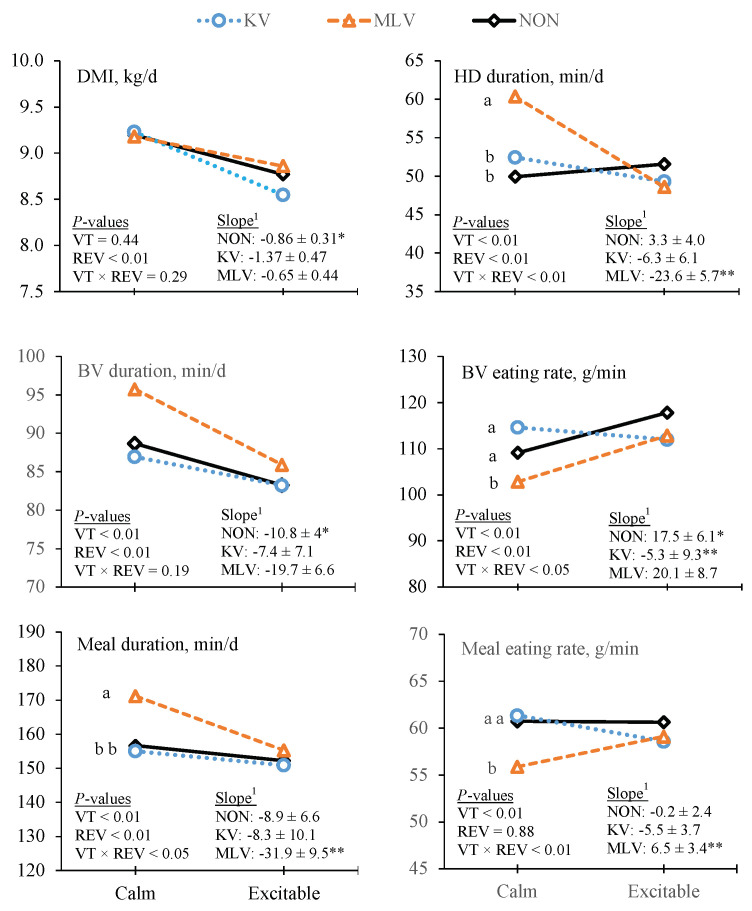
Effects of vaccine treatment (VT) and temperament on DMI and feeding behavior traits. ^1^ Slope = REV covariate ± SE for each VT. * The indicates that the slope for non-vaccinated (NON; *n* = 118) steers differed from zero (*p* < 0.01). ** This indicates that the slope of killed vaccine (KV; *n* = 119) or modified live vaccine (MLV; *n* = 123) steers differed (*p* < 0.05) from the slope of NON steers ^a,b^ This indicates a difference between subclass means at *p* < 0.05.

**Figure 4 animals-11-02133-f004:**
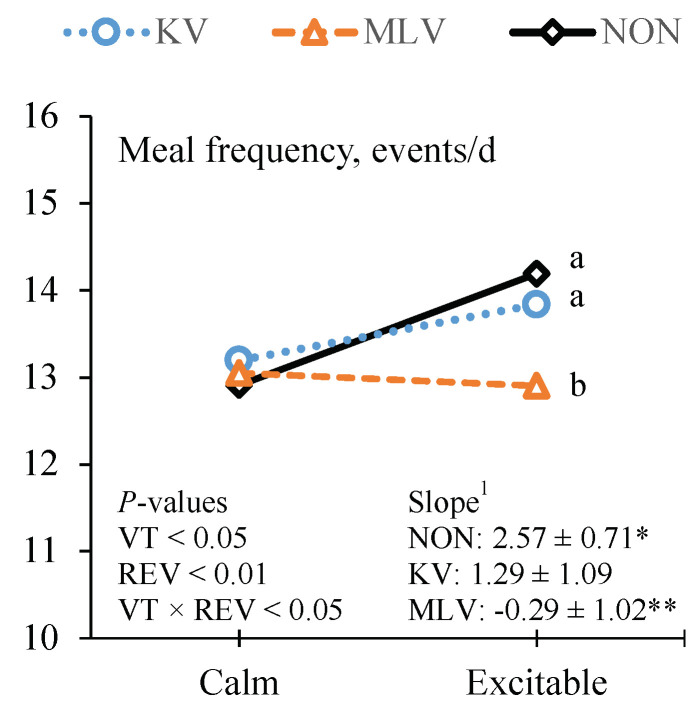
Effects of vaccine treatment (VT) and temperament on meal frequency. ^1^ Slope = REV covariate ± SE for each VT. * This indicates that the slope for non-vaccinated (NON; *n* = 118) steers differed from zero (*p* < 0.01). ** This indicates that the slope of killed vaccine (KV; *n* = 119) or modified live vaccine (MLV; *n* = 123) steers differed (*p* < 0.05) from the slope of NON steers. ^a,b^ This indicates differences (*p* < 0.05) between subclass means.

**Table 1 animals-11-02133-t001:** Definition of feeding behavior traits used in the study.

Trait	Description
Bunk visit (BV) frequency, events/d	Number of BV events for each day
BV duration, min/d	Sum of the lengths of all BV events recorded each day
BV eating rate, g/min	Daily dry matter intake divided by the daily BV duration
Head down duration (HD), min/d	Number of electronic identication recordings each day multiplied by the read rate of the GrowSafe system
Meal frequency, events/d	Number of meal events for each day
Meal duration, min/d	Sum of the lengths of all meal events recorded each day
Meal eating rate, g/min	Daily DMI divided by the daily meal duration
BV/meal (BVM), events/meal	BV frequency divided by meal frequency
Time to bunk (TTB), min/d	Length of interval between feed delivery and the first BV event

**Table 2 animals-11-02133-t002:** Effects of vaccine treatment and experimental period on DMI, performance and feeding behavior traits in response to BVDV challenge.

Trait	Vaccine Treatment (VT) ^1^				Experimental Period (EP)					*p*-Values			
	NON	KV	MLV	SE ^2^	1	2	3	4	SE ^2^	VT	EP	REV ^3^	VT × REV
DMI, kg/d *	8.98	8.89	9.02	0.11	9.53 ^a^	8.01 ^c^	9.02 ^b^	9.31 ^a^	0.12	0.44	<0.01	<0.01	0.29
ADG, kg/d	1.11	1.08	1.15	0.04	0.94 ^c^	0.68 ^d^	1.58 ^a^	1.24 ^b^	0.05	0.25	<0.01	0.08	0.29
G:F	0.122	0.121	0.129	0.005	0.100 ^c^	0.080 ^d^	0.178 ^a^	0.136 ^b^	0.005	0.20	<0.01	0.68	0.32
BV frequency, events/d ^4^	69.6	69.0	70.1	1.2	84.4 ^a^	63.6 ^c^	67.9 ^b^	62.4 ^c^	1.4	0.62	<0.01	0.36	0.08
BV duration, min/d	85.9 ^b^	85.1 ^b^	90.9 ^a^	1.7	93.4 ^a^	79.1 ^c^	87.2 ^b^	89.5 ^b^	1.9	<0.01	<0.01	<0.01	0.19
BV eating rate, g/min	113.4 ^a^	113.3 ^a^	107.8 ^b^	2.2	111.0	109.7	113.2	112.2	2.6	<0.01	0.55	<0.01	<0.05
HD duration, min/d ^4^	50.7 ^b^	50.9 ^b^	54.5 ^a^	1.5	57.1 ^a^	47.3 ^c^	51.9 ^b^	51.9 ^b^	1.7	<0.01	<0.01	<0.01	<0.01
Meal frequency, events/d	13.5 ^a^	13.5 ^a^	12.9 ^b^	0.3	15.2 ^a^	12.9 ^b^	13.2 ^b^	12.0 ^c^	0.3	<0.05	<0.01	<0.01	<0.05
Meal duration, min/d	154.5 ^b^	152.9 ^b^	163.3 ^a^	2.3	177.2 ^a^	139.1 ^c^	156.7 ^b^	154.6 ^b^	2.6	<0.01	<0.01	<0.01	<0.05
Meal eating rate, g/min	60.7 ^a^	59.9 ^a^	57.5 ^b^	0.9	56.0 ^c^	59.8 ^b^	59.5 ^b^	62.1 ^a^	1.0	<0.01	<0.01	0.88	<0.01
BVM, BV events/meal ^4^	5.44 ^b^	5.28 ^b^	5.67 ^a^	0.12	5.89 ^a^	5.12 ^c^	5.41 ^b^	5.44 ^b^	0.13	<0.01	<0.01	<0.05	0.97
TTB, min/d ^4^	28.9	28.7	28.3	1.6	24.6 ^b^	39.1 ^a^	24.2 ^b^	26.6 ^b^	1.8	0.91	<0.01	<0.01	0.11

^a–d^ Means within a row with different superscripts differ (*p* < 0.05). * A vaccine treatment (VT) × experimental period (EP) interaction was detected (*p* < 0.05). ^1^ Vaccine treatments (VT) include non-vaccinated (NON; *n* = 118), killed (KV; *n* = 119) and modified-live vaccine (MLV; *n* = 123). ^2^ SE of the mean difference. ^3^ REV = average relative exit velocity (Days 0 and 14) was utilized as a covariate. ^4^ BV = bunk visits; HD = head-down duration, BVM = bunk visits per meal  (BV frequency÷meal frequency), TTB = time to bunk.

**Table 3 animals-11-02133-t003:** Covariate regression slopes of relative exit velocity for DMI, performance and feeding behavior traits.

Trait ^2^	Intercept	Slope	SE	Temperament ^1^		*p*-Value	
				Calm	Excitable	REV	VT ^3^ × REV
DMI, kg/d	8.96	−0.94	0.19	9.19	8.73	<0.01	0.29
ADG, kg/d	1.11	−0.11	0.08	1.14	1.08	0.08	0.29
G:F	0.124	−0.002	0.009	0.125	0.124	0.68	0.32
BV frequency, events/d ^2^	69.5	1.97	2.22	69.0	69.9	0.36	0.08
BV duration, min/d	87.3	−12.7	2.81	90.5	84.1	<0.01	0.19
BV eating rate, g/min	111.3	11.6	3.69	108.4	114.2	<0.01	<0.05
HD duration, min/d ^2^	51.8	−8.38	2.44	53.9	49.7	<0.01	<0.01
Meal frequency, events/d	13.3	1.16	0.43	13.0	13.6	<0.01	<0.05
Meal duration, min/d	156.8	−16.2	4.04	160.9	152.8	<0.01	<0.05
Meal eating rate, g/min	59.4	0.24	1.46	59.3	59.5	0.88	<0.01
BVM, BV events/meal ^2^	5.47	−0.42	0.19	5.58	5.37	<0.05	0.97
TTB, min/d ^2^	28.8	7.45	2.61	26.9	30.7	<0.01	0.11

^1^ Temperament classification = mean relative exit velocity (REV) ± 1 SD. ^2^ BV = bunk visits; HD = head-down duration; BVM = bunk visits per meal, TTB = time to bunk. ^3^ VT = vaccine treatments, which include a modified live vaccine (MLV; *n* = 123), non-vaccinated (NON; *n* = 118) and a killed vaccine (KV; *n* = 119).

## Data Availability

The data presented in this study are available on request from the corresponding author. The data are not publicly available due to future intended analyses.

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
