# Peer review of "Effects of Multivalent BRD Vaccine Treatment and Temperament on Performance and Feeding Behavior Responses to a BVDV1b Challenge in Beef Steers"

_animals, 2021, doi:10.3390/ani11072133_

Round 1

Reviewer 1 Report

General comment: The authors presented an interesting and original work concerning to the effects of vaccine treatments for bovine respiratory disease on feed intake, performance and feeding behavior.

In a general way, the manuscript is well written.

Title: The title is too long. It should be short, concise, and informative. The use of abbreviations should be avoided.

Simple Summary: The abbreviation for Bovine Respiratory Disease (BRD) should be defined in the sentence: “Bovine respiratory disease threatens cattle production and welfare globally.”

Summary: The full name should be added before the abbreviation “DMI”.

The keywords should be different from those used in the title.

Introduction: It is clear, complete, and informative. The objective was clearly stated.

The abbreviations “MLV” and “KV” should be added to the following sentence: “In general, modified-live vaccines have been shown to elicit more robust and longer lasting immune responses 55 [9, 10] compared to killed vaccines.”

The abbreviation “BVDV” should be added to the last sentence: “… performance, and feeding behavior responses following a bovine viral diarrhea 70 virus challenge in growing beef steers.”

Materials and Methods:

In a general way, the Materials and Methods are well described.

Was the study approved by an Ethics Committee?

The abbreviations “BVDV” was previously defined in the Introduction. Please delete the full name and use only the abbreviation in the “Animal and Experimental design” (line 89).

Results:

The Results are clearly presented and supported by the Tables and Figures.

Conclusions:

The Conclusions are in accordance with the Results.

Recommendation:

The manuscript should be accepted after a Minor revision.

Reviewer 2 Report

This manuscript by Smith and the colleague have evaluated the eEffects of multivalent BRD vaccine treatment and temperament on performance and feeding behavior responses to a BVDV1b challenge in beef steers. The experimental was well designed and data was well presented. While the following changes could improve the quality of the paper.

  1. Line 33, correct “BW” to “Body weight”;
  2. Line 74-76, please provided the approved No. for this trial;
  3. Line 77, correct “N = 360” to “n = 360”;
  4. Table 1, please add the full name of “DMI” and “EID” in the table. Meanwhile, “Bunk visit (BV)” seems not the right way of writing in the Table. You can either use the full name or used the abbreviation in the table and add the full name in the footnote of the table.
  5. Table 2, please do not use the “bold” for the values with significant changes in the table.
  6. Figure 2, how many replicates have been used for the analysis? If there has siginificant, please used the different letter to express the changes. There are two independent figures in the Figure 2, please use A and B to defined them. The upper figure, please add another “b” for one of the treatment. Additionally, please added the standard error or deviation bar for each values in the figure.
  7. Table 3, please do not use the “bold” for the values with significant changes in the table. Meanwhile, adding the replicates (n = ?) in the footnote.
  8. Figure 3, please added the standard error or deviation bar for each values in the figure. Meanwhile, adding the replicates (n = ?). Additionally, label the different letters (A, B, C, D…) for each independent figures.
